Manuscript prepared for Atmos. Meas. Tech. with version 2014/09/16 7.15 Copernicus papers of the LATEX class copernicus.cls. Date: 27 September 2018

# Lidar temperature series in the middle atmosphere as a reference data set. Part B: Assessment of temperature observations from MLS/Aura and SABER/TIMED satellites

Robin Wing<sup>1</sup>, Alain Hauchecorne<sup>1</sup>, Philippe Keckhut<sup>1</sup>, Sophie Godin-Beekmann<sup>1</sup>, Sergey Khaykin<sup>1</sup>, and Emily M. McCullough<sup>2</sup>

<sup>1</sup>LATMOS/IPSL, UVSQ Université Paris-Saclay, Sorbonne Université, CNRS, Guyancourt, France <sup>2</sup>Department of Physics and Atmospheric Science, Dalhousie University, Halifax, Canada

Correspondence to: Robin Wing (robin.wing@latmos.ipsl.fr)

**Abstract.** We have compared 2433 nights of Rayleigh lidar temperatures measured at L'Observatoire de Haute Provence (OHP) with co-located temperature measurements from the Microwave Limb Sounder (MLS) and the Sounding of the Atmosphere by Broadband Emission Radiometry instrument (SABER). The comparisons were conducted using data from January 2002 to March 2018 in

- the geographic region around the observatory (43.93° N, 5.71° E). We have found systematic differences between the temperatures measured from the ground based lidar and those measured from the satellites which suggest non-linear distortions in the satellite altitude retrievals. We see a winter stratopause cold bias in the satellite measurements with respect to the lidar (-6 K for SABER and -17 K for MLS), a summer mesospheric warm bias (6 K near 60 km), and a vertically structured bias
- for MLS (-4 to 4 K). We have corrected the stratopause height of the satellite measurements using the lidar temperatures and have seen an improvement in the comparison. The winter relative cold bias between the lidar and SABER has been reduced to 1 K in both the stratosphere and mesosphere and the summer mesospheric warm bias is reduced to 2 K. Stratopause altitude corrections have reduced the relative cold bias between the lidar and MLS by 4 K in the early autumn and late spring but were
- unable to address the vertical artifacts in the MLS temperature profiles.

# 1 Introduction

Satellite atmospheric measurements are vital for providing global assessments of long term atmospheric temperature trends. However, particular care must be taken to validate each new satellite as well as provide periodic ground checks for the entire instrument lifetime in order to counter drifts in

calibration and local measurement time (Wuebbles et al., 2016). Changes in satellite measurements 20 can occur over the course of a mission due to instrument degradation, calibration uncertainties, orbit changes, and errors/assumptions in the forward model parameters. Additionally, most mission planning agencies have guidelines which require that satellite programs conduct formal validation studies to ensure accuracy and stability of the measurements (Council, 2007).

#### 1.1 Lidar as a Validation Tool 25

Rayleigh lidar remote sounding of atmospheric density and temperature is an excellent tool for use in validating satellite measurements over a specified geographic area and vertical range. Lidars can make routine high resolution measurements over a large portion of the middle atmosphere in regions which are notoriously difficult for other techniques to measure routinely or precisely. There are three key strengths in the Rayleigh lidar technique which set it apart from other atmospheric sounders. First is the ability to retrieve an absolute temperature profile from a measured relative density profile with very high spatio-temporal accuracy and precision. Second, lidars measure range by measuring the time required for a backscattered photon to return to the station and be recorded by the photon counting electronics. The current OHP lidar uses a Licel digital recorder and has

- a sampling 40 MHz which corresponds to a vertical resolution of 7.5 m. The uncertainty on the sampling rate is negligible however, there is the possibility of trigger delay and jitter in the counting electronics of 50  $\pm$  12.5 ns Licel (2018) contributing a maximum possible uncertainty of 18.25  $\pm$  3.25 m in the raw lidar measurement. This error is constant with altitude which allows us to sample the upper middle atmosphere with the same range resolved confidence as the lower middle
- atmosphere and troposphere. Third, as a benefit of active remote sensing raw lidar measurements don't suffer from vertical distortion in the altitude vector. Each altitude level in a lidar measurement corresponds to an independent collection of backscattered photons which are returning at a defined time from a given altitude range. In contrast, passive remote sensors such as limb scanning satellites can suffer biases at high altitudes due to: radiometric and spectral calibration, field of view and
- antenna transmission efficiency, satellite pointing uncertainty, as well as biases introduced by the 45 forward model (Schwartz et al., 2008). Additionally, many satellites like MLS are optimized for tropospheric and lower stratospheric measurements and conduct faster scans with fewer channels at higher altitudes (Livesey et al., 2006). These different biases can exist simultaneously in both the retrievals of temperature and pressure and can be considered, in part, as distortions in the altitude vector when compared to lidar measurements.

30

# 1.2 Previous Lidar-Satellite Temperature Studies

Previous studies comparing ground based lidar and satellite measurements of temperature have often used Sodium Resonance lidars to compare the lidar derived neutral temperature between 85 km and 105 km to satellite temperatures in the mesopause region. Studies of this sort have generally shown

good agreement between ground and satellite observations (Xu et al., 2006). Due to the strength of Na lidars in the upper mesosphere they naturally lend themselves well to studies of tides and wave breaking dynamics.

Coincident with this work (Dawkins et al., 2018) submitted a comparison of temperature profiles from 9 different metal resonance lidars with temperature profiles from SABER from 75 to 105 km.

- At all sites they found that SABER temperatures were cooler than the lidar temperatures by -9.9 (±9.7) K at 80 km. The study used coincidence criteria of ±15° longitude, ±5° latitude, and ±30 minutes between the lidar and satellite profiles. A weak and unexplained mesospheric summer bias was also reported. In the supplemental material to (Dawkins et al., 2018) a sensitivity study was done for SABER overpasses as a function of season and size of the co-location area. They found no significant differences between a co-location area with a longitudinal size of ±5° and ±15°.
  - A study by (Yuan et al., 2010) compared Na lidar and SABER temperatures in the context of a 6 year tidal analysis. They found semiannual disagreements in the tidal amplitude around the spring and autumn equinoxes with a maximum difference of 12 K near 90 km occurring in February. Several explanations and partial corrections were offered but the phenomenon is robust and the authors con-
- cluded that further study was required to fully resolve the temperature discrepancy. Studies have also been done comparing temperatures calculated from the Rayleigh lidar technique and those derived from SABER and MLS observations. (Taori et al., 2011, 2012a, b) comprise an excellence series of publications using multiple instruments to measure the atmospheric temperature from 40 km to 100 km. These works found good agreement between the lidar and SABER up to 65 km and signif-
- icant initialization errors in the lidar of up to 25 K near 90 km. We have partially accounted for this initialization induced lidar warm bias in the companion paper (Wing et al., 2018a). Our work here offers two improvements on these three publications. Firstly, we have not focused as much on case studies but rather on the statistics of nearly a decade of lidar-satellite inter-comparisons. Secondly, we have conducted our comparisons on a 1 km grid in an effort to match small scale features in the
- temperature profiles.

A good lidar to satellite temperature comparison was done by (Siva Kumar et al., 2003) using 240 nights of lidar temperatures, temperatures from UARS, and model temperatures from CIRA-86 and MSIS-90. They compared monthly and seasonal averages and found significant semiannual temperature anomalies in the region of 45 – 50 km in February-March and September-October as

- well as initialization related biases above 70 km. A second study by the same authors compared 14 years of monthly average lidar temperatures to temperatures from the satellites SABER, HALOE, COSMIC, and CHAMP (Sivakumar et al., 2011). As with the previous study temperature anomalies of 3-5 K were identified in the region near the stratopause. The differences were attributed to monthly averaging and slight differences in measurement time and location of the lidar and satellites. The
- approach employed in our work is to make comparisons of nightly averages and then study the

monthly median of the temperature differences – an approach which will allow for finer temporal precision.

Another study which compares 120 nights of Rayleigh lidar temperatures measured over Beijing to temperatures from SABER over the course of one year found good agreement between monthly average temperature profiles (Yue et al., 2014). This study found winter time temperature anomalies

- average temperature profiles (Yue et al., 2014). This study found winter time temperature anomalies in the stratopause region and attempted to account for these features by fitting an annual, semiannual, and 3 month sinusoid to the data. The objective of our study is similar to that of Yue et al. (2014) insofar as we are interested in the time evolution of lidar-satellite temperature comparisons and identifying potential seasonal or decadal trends. However, we are seeking to make nightly tem-
- perature comparisons between lidar and two satellites, SABER and MLS, over multiple years without assuming large contributions from an Annual Oscillation (AO) or its harmonics. Our study uses more than 9 times as many coincident measurements and spans the entire SABER data record.

Further study of seasonal temperature anomalies between ground based lidar and SABER was done by (Dou et al., 2009) comparing 2332 nights of lidar data from 6 different sites in the Network

- for the Detection of Composition Change (NDACC) to zonally averaged temperature profiles from SABER. This study found a 2-5 K systematic bias in the stratopause region and concluded that this result may be due to either a bias in SABER, tidal aliasing, or sporadic aerosols. As well the study found systematic temperature differences in the upper mesosphere which were attributed to tidal aliasing, bias in the SABER temperature retrieval, or temperature differences due to the AO. In our
- work we use a smaller geographic window and not a zonal average temperature to compare more truly co-incident measurements. As well we limit the time difference between the lidar and satellite measurements to minimize possible tidal contributions.

# **1.3** Alternative Measurement Techniques

Other current measurement techniques for atmospheric temperature in this region of the atmosphere 115 include:

a) Rocketsondes were used during the early satellite era to make in situ measurements of the middle atmosphere but this technique has many well known limitations and requires large corrections and uncertainties in the upper mesosphere (Johnson and Gelman, 1985).

b) Meteor radar techniques provide an estimation of the temperature at 90 km and can operate
on a near continuous basis but they require several a priori assumptions and must be calibrated with data from an independent source (Meek et al., 2013)

c) Satellites, like MLS and SABER provide globally distributed temperature measurements at several pressure levels throughout the vertical atmospheric column (Waters et al., 2006) (Mertens et al., 2001). Satellite based measurements provide a very good global view of the Earth's middle at-

125 mosphere but can suffer from calibration errors, temporal coverage gaps, and problems with vertical resolution.

d) OH airglow imagers (Pautet et al., 2014) provide high spatio-temporal resolution 2D images of temperature perturbations derived from OH emissions near 87 km. These instruments can provide excellent wide field of view measurements over a geographic area but cannot yield vertical profiles of temperature.

e) Ground-based resonance doppler and Boltzmann lidars can derive temperatures from sodium, iron, and other meteoric metal layers in the upper mesosphere and lower thermosphere (80 - 115 km) (Chu et al., 2002). These techniques are not only useful in deriving temperature profiles but are also well situated for studies of other middle atmospheric phenomena such as gravity waves

and noctilucent clouds. These lidars are restricted to measuring in the altitude band defined by the distribution of each metallic layer.

Considered together, this suite of remote sensing techniques can provide a comprehensive view of the middle atmosphere. The inclusion of Rayleigh lidar data into multi-sensor studies of the middle atmosphere provides an important local ground truthing perspective which helps to refine the global view offered by other techniques.

#### 1.4 Outline of this Work

In this work we give a brief description of the instruments involved in the study (Sect. 2), a definition of the geographic area under consideration, and several criteria for determining coincidence between lidar and satellite measurement profiles (Sect. 3). In Sect. 4 we directly compare temperature profiles

from MLS and SABER to the lidar temperatures and show a monthly median difference climatology and note several systematic differences. Section 5 details a procedure to correct the satellite temperature profiles based on the height of the stratopause in the lidar data. Finally, Sect. 6 shows an improved lidar-satellite monthly median difference climatology based on the altitude corrected satellite data.

#### 150 2 Instrumentation

The Observatoire de Haute Provence (OHP) Rayleigh lidars have been in operation in southern France since 1978 and routinely produce nightly average temperature profiles of the upper stratosphere and lower mesosphere. The details of the Rayleigh lidar algorithm and the OHP lidar specifications are presented in the companion publication (Wing et al., 2018a).

- SABER is a broadband radiometer aboard NASA's TIMED (Thermosphere Ionosphere Mesosphere Energetics Dynamics) satellite and makes temperature measurements based on CO<sub>2</sub> limb radiances from 20 km to 120 km. SABER has a vertical resolution of 2 km and random temperature errors of less than 0.5 K below 55 km, 1 K at 70 km, and 5 K at 100 km (Remsberg et al., 2008). TIMED does not have a sun synchronous orbit and does not pass though our OHP comparison area
- at a fixed local time. This makes finding temporally coincident measurements with the lidar rela-

tively easy. We are using version 2.0 of the published SABER temperatures. Further information for SABER/TIMED can be found in (Mertens et al., 2001).

MLS is an microwave spectrometer aboard the Aura satellite and makes temperature measurements based on emissions from  $O_2$ . Further information can be found in (Waters et al., 2006). MLS

vertical averaging kernels have a full-width-half maximum of 8 km at 30 km, 9 km at 45 km, and 14 km, at 80 km and a temperature resolution which goes from 1.4 K near 30 km to 3.5 K above 80 km (Schwartz et al., 2008). We are using version 4.0 of the published MLS temperatures. MLS is a sun synchronous satellite which passes the equator around 1h45 UTC and is generally temporally coincident with the last hour or so of lidar measurements.

#### 170 3 Comparison Parameters

Defining coincident measurements between satellites and lidars can be difficult due to temporal and spatial offsets, differences in viewing geometry, and different approaches to smoothing. Studies such as García-Comas et al. (2014) have defined short time windows over a 1000 km square surrounding the observatory as sufficient for coincidence while others such as (Yue et al., 2014) have chosen to approach the problem by looking at monthly averages over a much narrower latitude band.

For this study we wanted to compare to satellite profiles geographically near the lidar to minimize latitudinal variations in the temperature and within a small time frame to minimize the contribution of tides, tidal harmonics, and gravity wave effects. This desire for close spatio-temporal matching was balanced against the need for a sufficiently large number of comparisons as to produce results

- which are statistically significant and useful. Ultimately, we decided on a geographic window of  $\pm 4^{\circ}$  latitude and  $\pm 15^{\circ}$  longitude similar to the analysis done by (Dou et al., 2009). We reasoned that the UMLT (Upper Mesosphere and Lower Thermosphere) structure would vary with latitude to a greater degree than with longitude and that the longitudinal separation between consecutive SABER satellite passes gives a natural bound on the longitude. The contemporaneous work by (Dawkins
- et al., 2018) includes a sensitivity study on the choice of longitudinal co-location limits. Their final choice for a spatial coincidence ( $\pm 5^{\circ}$  latitude ,  $\pm 15^{\circ}$  longitude) is comparable to our study which employs ( $\pm 4^{\circ}$  latitude ,  $\pm 15^{\circ}$  longitude). Figure 1 shows the geographic extent of our study.

Figure 1: Area defined for coincident measurements (40° N, 9° E) to (48° N, 21° E). L'Observatoire de Havte Provence in blue at (43.93° N, 5.71° E). (data: Google, 2017)

The minimum length of an OHP nightly lidar temperature measurement is four hours. We chose to use a  $\pm 4$  h window around the lidar measurement as the temporal limit for coincidence with a satel-

- lite pass. This gives us a roughly 12 h window centered around the middle of the lidar measurement. Our choice was influenced by a desire to minimize the effect of the 12 h tidal harmonic. Previous work comparing between satellites have been able to take advantage of daytime satellite overpasses and chose to work within a  $\pm 2h$  window(Hoppel et al., 2008). (French and Mulligan, 2010) conducted a comparison between an OH spectrometer (in conjunction with a sodium lidar) and SABER
- at  $\pm 15$  min and  $\pm 8$  h and found no significant difference. However, it must be noted that this study 195 was conducted at a latitude of 69° S and the comparison may not hold in the mid-latitudes.

#### 4 Temperature comparisons without considering vertical offset

Here we demonstrate the directly-calculated temperature biases between OHP and both SABER and MLS which are present before we carry out the adjustment for satellite altitude offsets which are discussed in Sect. 5. An example of all three temperature profiles for the night of the 25th of July 200 2012 is shown in Fig. 2. In this comparison the lidar profile was produced over 4 hours and has a vertical resolution of 150 m from 30 km to above 90 km. The large temperature uncertainty above 70 km is a result of the fine vertical resolution required to capture the mesospheric inversion layer present near 77 km.