# Peer review of "Lidar temperature series in the middle atmosphere as a reference data set. Part B: Assessment of temperature observations from MLS/Aura and SABER/TIMED satellites"

_Atmospheric Measurement Techniques, 2018_

## Referee Comment (RC1) · Anonymous Referee #1 · 23 May 2018

**1 General remarks**

The authors present a long-term comparison of upper stratospheric and mesospheric temperature profiles derived from lidar measurements at Haute Provence with results from the SABER and MLS satellite instruments. They show altitude shifts in the satellite derived temperature profiles, seasonally varying differences that are not understood, and overall long-term consistency.

I believe this is an important and generally well written paper, and well suited for publication in AMT after addressing the following short-comings.

**2 Detailed suggestions**

In the following, the numbers *x*, *y* refer to page *x* and line *y* of the manuscript.

1, 4: Why only to 2011? We are now in 2018! MLS and SABER and the OHP lidar are still working and providing temperature profiles. Please use the additional 6 years of data since 2012, and provide results that are much more meaningful.

At some point in the paper it is absolutely necessary to mention / show, how the lidar temperature analysis presented here relates to the temperature profiles published for many years in the NDACC database. Are there systematic differences between the two? If so, where and how big? Maybe even an additional plot.

2, 26-36 This does not connect well to the previous paragraph. Before you talked about satellites as primary instruments. Here, suddenly, you talk about alternatives to lidar. Please rework the entire introduction, so that there is a more logical flow.

2, around 45: Why not say that lidars measure altitude / range via measuring time, and that this is a very precise measurement with relative uncertainty of the order of  $10^{-6}$  (or whatever the electronics of the OHP lidar specify).

2, around 48: It would be good to give a reference for this claimed distortion of the altitude vector.

3, around 63: It would make sense to give pros and cons also for the airglow imagers, similar to what is done for the other techniques.

Also: Sodium and other metal layer lidars should also be introduced briefly in this context, including their pros and cons.

4, 90: Siva Kumar or Sivakumar. Many reference callouts, and many references are sloppy. They all need to be checked and corrected.

4, 94: Is it an "initialization problem" or "initialization related bias"? To me, problem seems the wrong word.

4, 117-118: Sentence seems to be broken / missing something.

5, 124: Are these the numbers that are relevant for this study? Seems to me that a usual temperature profile is acquired over at least 4 hours (page 6, line 168). It would make more sense to use the more relevant times and altitude resolutions of the retreived profiles here, not the ones of the underlying data acquisition.

5, 143: Drop "Other"?

Figure 2: Good figure. I would suggest, however, to also present average temperature profiles from lidar and MLS before Fig. 2. This will set the stage and help readers who do not have the average temperature profile in their head. It will also lead nicely into the vertical shifts discussed later.

Figure 2 caption: "show" should be "shown".

Figs. 3, 5, 11, 12: Color scale is missing.

7, 175: Could that not be checked, whether there is a bias coming from the initialization, e.g. by using MLS or SABER temperatures, or at least comparing them with the used initialization temperatures. I think more digging into this is required and would be a very important test for this paper.

7, 178: How do you know that lidars are exceptionally accurate there? I think this needs more explanation and / or a reference (e.g. Leblanc et al. AMT 2016). Or do you mean precise, which is easier to show than accurate? What is exceptional? 0.01 K? 0.1 K is typical for radiosondes at lower altitudes around 10 to 20 km, and would not be exceptional. Also, instead of "are" I would prefer "should be".

СЗ

Figure 10 caption: It would be good to say that the underlying color plots are the same as in Figs. 2 and 4.

Figs. 11, 12: It would be good to also show seasonal difference profiles, similar to Figs. 2, 4. This would be particularly good for showing the oscillations in the lidar - MLS differences.

17, around 240: I am missing plots and a discussion of the time-altitude evolution of the lidar - SABER and lidar - MLS differences after the altitude shift corrections have been applied (Similar to Figs. 2 and 4). In particular it would be good discuss whether there are long-term drifts in these differences, or whether all instruments seem stable over time and thus usable for the temperature trend detection outlined in the introduction. Probably there needs to be some analysis looking into possible long-term trends in these differences. As mentioned before, this should include data up to 2018.

17, 258: The "why" for this needs to be discussed, not just shrugged off. Is it really background correction? Is it noise, i.e. are noisier profiles biased more (this could easily tested by comparing e.g. four 1 hour profiles with the corresponding 4 hour profile.) Or is it initialization temperature (test how much it would have to be changed to get rid of the bias, and how consistent that is with e.g. SABER, MLS at high altitudes).

18, 275: Other things that come to mind here, and should be mentioned, are multiple scattering effects not considered in the single scattering lidar equation. This could result in enhanced return signals at lower altitudes, which pretends too high density and too cold temperature. Also, smaller rotational Raman bandwidth from the light scattered in colder regions (lower stratosphere) results in enhanced effective system transmission for those altitudes, also pretend too high density and too cold temperatures (She at al. 200x, Whiteman et al. 200x). Also: Is ozone absorption accounted for correctly? I think it would be important to have some numbers for the possible magnitude for all these effects (including the ones currently in the manuscript), for the OHP lidar configuration.

19, 327: Remove "located". A "spatial" verb seems wrong in this temporal context.

19, 331-333: I did not see much discussion of accuracy and precision in this paper (e.g. hardly any standard deviations, uncertainty estimates and their checks.). Largely, the paper looks only at satellite - lidar bias and its temporal evolution. Therefore, I would rather say that the lidar provide good temperature measurements that are consistent with SABER and MLS over a decade (decades only if data up to 2017 or 2018 are analyzed, as suggested at the beginning.

The references are rather sloppy and need to be checked carefully.

Like many manuscripts, this one would also benefit from reducing redundancies and improving conciseness. I realize that addressing my remarks above will initially tend to make the paper longer. However, I would urge the authors to go through the paper again carefully and remove redundancies and repetitions where possible. As mentioned, this is basically a good and important paper, and should be made as readable as possible.

---

## Author Response (AR1)

**Response Lidar temperature series in the middle atmosphere as a reference data set. Part B: Assessment of temperature observations from MLS/Aura and SABER/TIMED satellites Referee #1**

In the following, the numbers x, y refer to page x and line y of the manuscript.
* * *
* * *
1, 4: Why only to 2011? We are now in 2018! MLS and SABER and the OHP lidar are still working and providing temperature profiles. Please use the additional 6 years of data since 2012, and provide results that are much more meaningful.

**There were two reasons I ended the analysis at 2011:**
1) **The LTA system underwent significant system upgrades in 2011 and as a result has a few data gaps.  In part A of the article I identified a 20 year period for comparison where both lidars remained relatively unchanged.  After establishing the lidars as a consistent benchmark measurement I wanted to use the same time period in the satellite comparison.**
2) **There were several periods after and during system development and change where the lidar data cadence or quality was well below average and I rejected the profiles as candidates for this study.**

**I have extended the analysis from 2011 to 2018 by using the temperature profiles from LiO3S (which were validated in Part A) to fill in the gaps in the LTA data record. It is important to note that LiO3S is a stratospheric ozone lidar and was not designed to measure temperatures high into the mesosphere.  As I result I have increased the vertical integration for these profiles.**

**Text has been added and modified throughout the article to accommodate these changes.**

**Figures 3, 4, 5, 6, 10 11, 12, 13 have been updated**
* * *
* * *
At some point in the paper it is absolutely necessary to mention / show, how the lidar temperature analysis presented here relates to the temperature profiles published for many years in the NDACC database. Are there systematic differences between the two? If so, where and how big? Maybe even an additional plot.

**This is shown and discussed in Figure 11 of the companion article (Wing et al., 2018a)**

[Figure]

The black zero line are temperatures produced by the NDACC lidar algorithm for LTA, coloured lines are the median ensemble temperature differences for the algorithm presented in the companion article (green), the ozone lidar LiO3S (orange), MSIS-90 (magenta), SABER (blue), and MLS (red). Blue shaded area is the variance of all SABER-lidar comparisons and is given to illustrate the scale of geophysical variations. The systematic differences between the NDACC and modified temperature algorithm (black and green) are negligible below 70 km, -4 K at 80 km, and -20 K at 90 km. The change in character above 84 km is mainly due to species specific Rayleigh backscatter correction and changes to the gravity vector.
* * *
2, 26-36 This does not connect well to the previous paragraph. Before you talked about satellites as primary instruments. Here, suddenly, you talk about alternatives to lidar. Please rework the entire introduction, so that there is a more logical flow.

Removed sentence:
**In this work we will show the value of our improved Rayleigh lidar temperature profiles, described in (Wing et al., 2018a), as a validation tool in the middle atmosphere .**

Reworked the introduction into the following format:
**Section 1.1 Rayleigh lidar as a validation tool**
**Section 1.2 Precious lidar-satellite studies**

**Section 1.3 Alternative validation techniques**
**Section 1.4 Outline**
* * *
* * *
2, around 45: Why not say that lidars measure altitude / range via measuring time, and that this is a very precise measurement with relative uncertainty of the order of 10 (or whatever the electronics of the OHP lidar specify).

Replaced this  sentence with:
 **Second, lidars measure range by measuring the time required for a backscattered photon to return to the station and be recorded by the photon counting electronics. The current OHP lidar uses a  Licel digital recorder and has a sampling 40 MHz which corresponds to a vertical resolution of 7.5 m.  The uncertainty on the sampling rate is negligible however, there is the possibility of trigger delay and jitter in the counting electronics of 50 ± 12.5 ns \cite{Licel_manual} contributing a maximum possible uncertainty of 18.25 ± 3.25 m in the raw lidar measurement.  This error is constant with altitude which allows us to sample the upper middle atmosphere with the same range resolved confidence as the lower middle atmosphere and troposphere.**
* * *
* * *
2, around 48: It would be good to give a reference for this claimed distortion of the altitude vector.

Replaced this  sentence with:
**Third, as a benefit of active remote sensing raw lidar measurements don't suffer from vertical distortion in the altitude vector.  Each altitude level in a lidar measurement corresponds to an independent collection of backscattered photons which are returning at a defined time from a given altitude range.  In contrast, passive remote sensors such as limb scanning satellites can suffer biases at high altitudes due to: radiometric and spectral calibration, field of view and antenna transmission efficiency, satellite pointing uncertainty, as well as biases introduced by the forward model \citep{schwartz_2008_MLS_validation}.  Additionally, many satellites like MLS are optimized for tropospheric and lower stratospheric measurements and conduct faster scans with fewer channels at higher altitudes \citep{Livesey et al 2006}. These different biases can exist simultaneously in both the retrievals of temperature and pressure and can considered, in part, as distortions in the altitude vector when compared to lidar measurements.**
* * *
* * *
3, around 63: It would make sense to give pros and cons also for the airglow imagers, similar to what is done for the other techniques.
Also: Sodium and other metal layer lidars should also be introduced briefly in this context, including their pros and cons.

Added sentence:
**These instruments can provide excellent wide field of view measurements over a geographic area but cannot yield vertical profiles of temperature.**

Added section:
**e) Ground-based resonance doppler and Boltzmann lidars can derive temperatures from sodium, iron, and other meteoric metal layers in the upper mesosphere and lower thermosphere (80 - 115 km) \citep{Fe_temperature_lidar_Chu}. These techniques are not only useful in deriving temperature profiles but are also well situated for studies of other middle atmospheric phenomena such as gravity waves and noctilucent clouds. These lidars are restricted to measuring in the altitude band defined by the distribution of each metallic layer.**
* * *
* * *
4, 90: Siva Kumar or Sivakumar. Many reference callouts, and many references are sloppy. They all need to be checked and corrected.

**The 2003 article is listed as V. Siva Kumar and the 2011 article as V. Sivakumar.**

**Because this is the format of the author's name in the original publications, I have retained that formatting here so that readers can located and access the correct journal articles in each case.**

**I export my references into BibTex directly from the journal websites.**
* * *
* * *
4, 94: Is it an "initialization problem" or "initialization related bias"? To me, problem seems the wrong word.

**Changed to initialization related bias.**
* * *
* * *
4, 117-118: Sentence seems to be broken / missing something.
The study also found found a systematic difference in the upper mesosphere which was attributed to tidal aliasing, bias in SABER or AO.

Changed to:
**As well the study found systematic temperature differences in the upper mesosphere which were attributed to tidal aliasing, bias in the SABER temperature retrieval, or temperature differences due to the AO.**
* * *
* * *
5, 124: Are these the numbers that are relevant for this study? Seems to me that a usual temperature profile is acquired over at least 4 hours (page 6, line 168). It would make more sense to use the more relevant times and altitude resolutions of the retrieved profiles here, not the ones of the underlying data acquisition.

Changed to:
**The Observatoire de Haute Provence (OHP) Rayleigh lidar has been in operation in southern France since 1978 and routinely produces nightly average temperature profiles of the upper stratosphere and lower mesosphere.**
* * *
* * *
5, 143: Drop "Other"?

**Done**
* * *
* * *
I would Figure 2: Good figure. suggest, however, to also present average temperature profiles from lidar and MLS before Fig. 2. This will set the stage and help readers who do not have the average temperature profile in their head. It will also lead nicely into the vertical shifts discussed later.

**I've added a new figure 2 with an example temperature profile for each instrument. Also added some text:**
**An example of all three temperature profiles for the night of the 25th of July 2012 is shown in figure \ref{fig:mil}.  In this comparison the lidar profile was produced over 4 hours and has a vertical resolution of 150 m from 30 km to above 90 km.  The large**

**temperature uncertainty above 70 km is a result of the fine vertical resolution required to capture the mesospheric inversion layer present near 77 km.**
* * *
Figure 2 caption: "show" should be "shown".

**Changed**
* * *
Figs. 3, 5, 11, 12: Color scale is missing.

**Added.**
* * *
7, 175: Could that not be checked, whether there is a bias coming from the initialization, e.g. by using MLS or SABER temperatures, or at least comparing them with the used initialization temperatures. I think more digging into this is required and would be a very important test for this paper.

**Yes the idea of initializing the lidar retrieval with an external temperature is a good one.**

**I had initially considered using OH airglow temperatures to initialize the lidar as well as satellite temperatures and then doing as you suggest and comparing back to the satellites. Unfortunately, this is not a test which I can perform in a timely manner. I started writing the codes to do this analysis and it quickly became clear that this project is not so straight forward. Ensuring that the gridding for the initialization point is correct, error propagation, in both temperature and altitude, for a lidar retrieval using three different instruments for seed pressure, and thinking about what kind of statistics are meaningful to use when comparing the lidar, as a function of satellite temperature and pressure, to the satellite would be both complicated and important. If circumstances permit I'd like to come back to this idea after I complete my thesis and write up the results in as a separate paper.**
* * *
7, 178: How do you know that lidars are exceptionally accurate there? I think this needs more explanation and / or a reference (e.g. Leblanc et al. AMT 2016). Or do

you mean precise, which is easier to show than accurate? What is exceptional? 0.01 K? 0.1 K is typical for radiosondes at lower altitudes around 10 to 20 km, and would not be exceptional. Also, instead of "are" I would prefer "should be".

**Point well taken.  I've softened the statement and added the relevant Leblanc citations**
Changed to:
… **a region where lidar uncertainties in both altitude and temperature  are well described \citep{leblanc_ndacc1} \citep{leblanc_ndacc3}**
* * *
* * *
Figure 10 caption: It would be good to say that the underlying color plots are the same as in Figs. 2 and 4.

**Done**
* * *
* * *
Figs. 11, 12: It would be good to also show seasonal difference profiles, similar to Figs. 2, 4. This would be particularly good for showing the oscillations in the lidar - MLS differences.

**Added Figure 14 to show the change in the ensemble plots for all temperature comparisons, summer comparisons, and winter comparisons.**

**I've also added text in support of the figure and to the discussion.**
* * *
* * *
17, around 240: I am missing plots and a discussion of the time-altitude evolution of the lidar - SABER and lidar - MLS differences after the altitude shift corrections have been applied (Similar to Figs. 2 and 4). In particular it would be good discuss whether there are long-term drifts in these differences, or whether all instruments seem stable over time and thus usable for the temperature trend detection outlined in the introduction. Probably there needs to be some analysis looking into possible long-term trends in these differences. As mentioned before, this should include data up to 2018.

**I have added figure 14 which shows the ensemble medians before and after correcting for stratopause height.**

**I intend to look at altitude dependant decadal temperature trends in my next article.**

**Cutting this article off with a discussion of seasonal variations seems like a good end point.  I have to think carefully about how to extract the seasonal component of the variation from any systematic change over the 16 year period.  As well I would like to discuss how best to resolve the disagreements between lidar and the satellites with someone from both the SABER and MLS team.**

**As noted in earlier in the response, the analysis now includes data up to March 2018.**
* * *
17, 258: The "why" for this needs to be discussed, not just shrugged off. Is it really background correction? Is it noise, i.e. are noisier profiles biased more (this could easily tested by comparing e.g. four 1 hour profiles with the corresponding 4 hour profile.) Or is it initialization temperature (test how much it would have to be changed to get rid of the bias, and how consistent that is with e.g. SABER, MLS at high altitudes).

**Added:**
**There still remains some residual systematic warm bias between the lidar satellite comparisons in this publication. Further work needs to be done on the problem of lidar initialization to fully address the effects of noise and a priori choice on high altitude Rayleigh lidar retrievals. However, we cannot discount the possibility that some of the remaining temperature difference is due to incorrect altitudes in the satellite data product.**

**Cited Wing2018A results regarding cooling due to noise filtration at the top of the lidar profile**
* * *
18, 275: Other things that come to mind here, and should be mentioned, are multiple scattering effects not considered in the single scattering lidar equation. This could result in enhanced return signals at lower altitudes, which pretends too high density and too cold temperature. Also, smaller rotational Raman bandwidth from the light scattered in colder regions (lower stratosphere) results in enhanced effective system transmission for those altitudes, also pretend too high density and too cold temperatures (She at al. 200x, Whiteman et al. 200x). Also: Is ozone absorption accounted for correctly? I think it would be important to have some numbers for the possible magnitude for all these effects (including the ones currently in the manuscript), for the OHP lidar configuration.

**We have discussed in Part A (Wing et al 2018a) our rationale for ignoring multiple scatter effects.**

Multiple scatter effects are negligible.  The probability of a photon backscattering is small, the probability of a photon backscattering twice is vanishingly small, and the probability of a twice backscattered photon being inside the lidar field of view (0.27 mrad see: Table 1 in Wing et al 2018a) is near zero.  Ignoring the multiple scatter terms in the lidar equation is standard practice in middle atmospheric studies.  As well any multiple scatter effects from water clouds in the troposphere would not be seen in the OHP Rayleigh lidar as the low gain channel is electronically blanked at 12 km and the high gain at 22 km.

OHP lidar has very narrow bandpass filters, either 1 nm for older measurements or 0.3 nm for recent years (Wing et al. 2018a). The rotational raman lines are outside of our bandpass.  We are currently working to develop and install a rotational Raman temperature channel for temperatures from the ground to 30 km. This is the current project of another PhD student.

An example O3 correction is shown below:

[Figure]
* * *
* * *
19, 327: Remove "located". A "spatial" verb seems wrong in this temporal context.

**Changed to 'found'**
* * *
* * *
19, 331-333: I did not see much discussion of accuracy and precision in this paper (e.g. hardly any standard deviations, uncertainty estimates and their checks.). Largely, the paper looks only at satellite - lidar bias and its temporal evolution. Therefore, I would

rather say that the lidar provide good temperature measurements that are consistent with SABER and MLS over a decade (decades only if data up to 2017 or 2018 are analyzed, as suggested at the beginning.

**Section 7 has been re-written. Closer attention was paid when using the words "accurate" and "precise"**
* * *
* * *
The references are rather sloppy and need to be checked carefully.
Like many manuscripts, this one would also benefit from reducing redundancies and improving conciseness. I realize that addressing my remarks above will initially tend to make the paper longer. However, I would urge the authors to go through the paper again carefully and remove redundancies and repetitions where possible. As mentioned, this is basically a good and important paper, and should be made as readable as possible

**We have double checked the references. Some (Sivakumar vs. Siva Kumar, for example) appear incorrect but in fact match the author names on the original publications.**

**We have endeavoured to reduce redundancies where possible, while also incorporating all sections which the reviewers sought to have added to the manuscript.**

**Response Lidar temperature series in the middle atmosphere as a reference data set. Part B: Assessment of temperature observations from MLS/Aura and SABER/TIMED satellites Referee #2**

The main weaknesses of the current manuscript are:
1) the short (and old) time span of the comparisons knowing that all 3 instruments in question are still operational today, and
2) there is little, or no, investigation of the differences that are not explained only by altitude shift. To this respect, I encourage the authors to invite the MLS and SABER temperature validation teams to provide their inputs (and possibly add them as co-authors)

**I have extended the analysis from 2011 to 2018 by using the temperature profiles from LiO3S (which were validated in Part A) to fill in the gaps in the LTA data record. It is important to note that LiO3S is a stratospheric ozone lidar and was not designed to measure temperatures high into the mesosphere. As I result I have increased the vertical integration for these profiles. Text has been added and modified throughout the article to accommodate these changes. Figures 3, 4, 5, 6, 10, 11, 12, 13 have been updated.**

**I have contacted both the SABER and MLS teams and offered co-authorship and to make any required changes to the article. Investigators either declined co-authorship, didn't positively indicate a desire for co-authorship, or didn't respond.**
* * *
I therefore suggest publication after major revisions, which: 1) Include a longer time period (e.g., 2004-2017) 2) Include inputs from MLS and SABER satellite teams 3) Include further investigation of the observed differences that may arise from lack of temporal and horizontal co-location

**1) I have extended the analysis from 2011 to 2018 by using the temperature profiles from LiO3S to fill in the LTA data record. See response to previous comment.**

**2) See response to previous comment regarding input from MLS and SABER teams.**

**3) Dawkins et al, 2018 was published last month and presents a systematic comparison of 9 lidar sites with SABER. In this article they show the effect of small variations in co-incidence criteria have little real difference on the comparison. They used similar horizontal co-location criteria to what I first presented here.**

**I have added a discussion of Dawkins et al, 2018 to this paper.**
* * *
Line 59-60: Schwartz et al, 2008 should be included here. Also, check more recent publications (for e.g., referring to GOZCARDS)

**Schwartz et al, 2008 does not compare MLS to a lidar but the reference has been included in the conclusion along with the following text:**

**"The results of this study will be useful for any future satellite validation studies in the style of (Schwartz et al. 2008) where lidar data could be used as a reference dataset.  In particular, lidar - satellite bias study results are useful for the ongoing NASA project "The Mesospheric and Upper Stratospheric Temperature and Related Datasets" (MUSTARD) which seeks to merge historic and ongoing satellite datasets."**
* * *
Figures 3 and 5: I suggest showing the temperature fields as well, at least for lidar, and preferably for both lidar and satellite. This way, differences on the 2D contour plots can perhaps be associated with specific temperature features

**Figure 2 has been added with an example of a nightly co-located temperature profile from the lidar, MLS and SABER.  The 2D temperature fields for the lidar and satellites are not particularly informative.  You can see the annual oscillation and some time periods where the lidar data was not so great (isolated periods after 2010 and in particular around 2015) but picking out particular features by eye is challenging.**

[Figure]

Lines 255-260: There is little quantitative discussion of the temperature uncertainties

throughout this manuscript. Although I understand there is a "Part 1" manuscript, a figure showing typical systematic, random and total uncertainties for lidar, MLS and SABER, as a function of altitude, would be very useful.

**Figure 2 has been added with nightly mean temperature and uncertainties. Part 1 of this paper has been modified to include a presentation of lidar uncertainties.**

Manuscript prepared for Atmos. Meas. Tech.
with version 2014/09/16 7.15 Copernicus papers of the LaTeX class copernicus.cls.
Date: 27 September 2018

**Lidar temperature series in the middle atmosphere as a reference data set. Part B: Assessment of temperature observations from MLS/Aura and SABER/TIMED satellites**

Robin Wing[1], Alain Hauchecorne[1], Philippe Keckhut[1], Sophie Godin-Beekmann[1], Sergey Khaykin[1], and Emily M. McCullough[2]

[1]LATMOS/IPSL, UVSQ Université Paris-Saclay, Sorbonne Université, CNRS, Guyancourt, France
[2]Department of Physics and Atmospheric Science, Dalhousie University, Halifax, Canada

*Correspondence to:* Robin Wing (robin.wing@latmos.ipsl.fr)

**Abstract.** We have compared  2433 nights of Rayleigh lidar temperatures measured at L'Observatoire de Haute Provence (OHP) with co-located temperature measurements from the Microwave Limb Sounder (MLS) and the Sounding of the Atmosphere by Broadband Emission Radiometry instrument (SABER). The comparisons were conducted using data from January 2002 to

5  March 2018 in the geographic region around the observatory (43.93° N, 5.71° E). We have found systematic differences between the temperatures measured from the ground based lidar and those measured from the satellites which suggest non-linear distortions in the satellite altitude retrievals. We see a winter stratopause cold bias in the satellite measurements with respect to the lidar (-6 K for SABER and  -17 K for MLS), a summer mesospheric warm bias ( 6 K near 60 km), and a

10  vertically structured bias for MLS ( -4 to 4 K). We have corrected the stratopause height of the satellite measurements using the lidar temperatures and have seen an improvement in the comparison. The winter  relative cold bias between the lidar and SABER has been  reduced to 1 K in both the stratosphere and mesosphere and the summer mesospheric warm bias is reduced  to 2 K. Stratopause altitude corrections have reduced the relative cold bias between

15  the lidar and MLS by 4 K in the early autumn and late spring but were unable to address the vertical artifacts in the MLS temperature profiles.

**1 Introduction**

Satellite atmospheric measurements are vital for providing global assessments of long term atmospheric temperature trends. However, particular care must be taken to validate each new satellite as

20  well as provide periodic ground checks for the entire instrument lifetime in order to counter drifts in calibration and local measurement time (Wuebbles et al., 2016). Changes in satellite measurements can occur over the course of a mission due to instrument degradation, calibration uncertainties, orbit changes, and errors/assumptions in the forward model parameters. Additionally, most mission planning agencies have guidelines which require that satellite programs conduct formal validation studies

25  to ensure accuracy and stability of the measurements (Council, 2007).

30  ~~with respect to a lidar measurement Sect. 1.3. In Sect. 1.2 we outline several previous lidar-satellite temperature comparison studies and contrast their methods to our study. Following the introduction is 
[revised manuscript text omitted]

[Figure]

Figure 2: Example co-located temperature profiles from the OHP lidar (green), SABER (blue), MLS (red), and MSIS (black).

**4.1  Comparison OHP Lidar and SABER**

240    From 2002 to  2018 there were 1100 coincident measurements of sufficient quality between OHP  lidars and SABER. Figure  3 (upper panel) shows the monthly median temperature differences between the lidar and SABER while Fig.  3 (lower panel) shows the mean seasonal temperature bias with altitude.

[Figure]

Figure 3:  Sixteen year systematic comparison of OHP  lidars and SABER temperatures. The monthly median temperature differences between the lidar and SABER are  shown in the upper panel. Red indicates that the lidar is warmer than SABER and blue that the lidar is colder. There are  1100 nights of coincident measurements in the colour plot. The lower panel is a seasonal ensemble of lidar minus SABER temperature differences. The summer (May, June, July, August) ensemble in red includes  306 nights of coincident measurements and the winter (November, December, January, February) ensemble in blue includes  397 nights of coincident measurements. Shaded errors represent 1 and 2 standard deviations.

Figure 3 upper panel contains the monthly median temperature differences between an OHP lidar
245   temperature profile and a SABER temperature profile. After 2010 there are several time periods where the Lidar Température et Aérosol (LTA) was not in routine operation or was in the process of being upgraded. To fill in these data gaps we have used temperature profiles derived from the ozone Differential Absorption Lidar (DIAL), also referred to as Lidar Ozone Stratosphère (LiO$_3$S), which is described and validated for temperature in (Wing et al., 2018a). Given that the main scientific
250   interest of LiO$_3$S is stratospheric ozone, the noise floor of the raw lidar signal occurs a lower altitude than for LTA for similar vertical integration. To produce temperature profiles which extend into the

mesosphere we use a coarser vertical resolution, a minimum altitude of 30 km, and often stop the temperature profile below 80 km if the temperature error becomes excessive.

The upper panel shows a relative warm bias for the  lidars with respect to SABER above 70 km. Discrepancies in this region are likely due to lidar initialization errors and background uncertainty which we have attempted to minimize in the companion publication (Wing et al., 2018a). There is also an evident seasonal relative warm bias in the winter stratosphere between 30 km and 50 km - a region where lidar  uncertainties in both altitude and temperature are well described  (Leblanc et al., 2016). The lower panel shows a very distinctive 'S' shape to the bias in both the winter and summer ensembles which is indicative of a vertical offset between the lidar and satellite measurements. The basic 'S' shape bias was identified in studies of synthetic lidar data as being due to vertical offsets between lidar instruments (Leblanc et al., 1998). Unfortunately, this offset is neither constant from night to night, nor constant with altitude as evidenced by the elongated and distorted nature of the 'S' shape.

If we bin all the temperature differences by month we can clearly see that there is a winter stratospheric warm bias below 45 km and a pronounced summer cold bias in the mesosphere between 50 and 70 km, as shown in Fig. 4.

[Figure]

Figure 4: Monthly median temperature difference between lidar and SABER temperature measurements. Red indicates regions where the lidar measures warmer temperatures than SABER and blue regions where the lidar measures colder temperatures than SABER.

**4.2 Comparison OHP lidar and MLS**

From 2004 to  2018 there were 1741 coincident measurements of sufficient quality between OHP lidars and MLS. Figure 5 (upper panel) shows the monthly median

temperature differences between the lidar and MLS while Fig. ??-5 (lower panel) shows the mean seasonal temperature bias with altitude.

As was the case with the lidar-SABER comparison, in the upper panel we see a lidar warm bias
275 above 70 km and a strong winter stratospheric warm bias near 4045 km. In this comparison the stratospheric warm bias appears to have a downward phase migration as the winter progresses. In the corresponding lower panel we see very pronounced summer time systematic differences which alternate from warm to cold throughout the stratosphere and mesosphere. The winter ensemble shows a very large lidar warm bias near the stratopause.

[Figure]

Figure 5: Six Fourteen year systematic comparison of OHP lidar and MLS temperatures. The monthly median temperature differences between the lidar lidars and MLS are show shown in the upper panel. There are 717 1741 nights of coincident measurements. The lower panel is a seasonal ensemble of lidar minus MLS temperature differences. The summer (May, June, July, August) ensemble in red includes 224 554 nights of coincident measurements and the winter (November, December, January, February) ensemble in red blue includes 269 653 nights of coincident measurements. Shaded errors represent 1 and 2 standard deviations.

280      Following the same procedure of binning lidar-MLS temperature differences by month we see a very pronounced downward phase progression of the winter stratospheric warm bias  from 45 km in January descending down to 40 km in February and March. Additionally, there is an evident layered cold bias in the summer stratosphere and mesosphere. The three layers appear near 37  km, 53 km, and 68 km in Fig. 6.

[Figure]

Figure 6: Monthly median temperature difference between lidar and MLS temperature measurements. Red indicates regions where the lidar measures warmer temperatures than MLS and blue regions where the lidar measures colder temperatures than MLS.

285 **5   Minimizing Temperature Difference Between Lidar and Satellites with a Vertical Offset**

We investigated a possible vertical offset between the lidar and satellite measurements to determine whether this could be contributing to the temperature biases seen in Sect. 4.

**5.1   Method to determine the vertical offset between measurements**

To match the two temperature profiles exactly in amplitude and altitude requires a unique altitude 290 dependent correction factor for each comparison. However, we can make a rough estimate of the average vertical offset between the two measurements by focusing on the region of the statopause which generally has a defined altitude and a clear structure. We used a simple least squares method to best estimate the vertical offset that would minimize the temperature differences between the lidar measurement and the satellite measurement. Two examples of this offset calculation for SABER are 295 shown in Fig. 7 and two examples for MLS are shown in Fig. 8.  The examples in these figures show nights where the lidar and satellite temperatures are in good agreement or can be brought into good agreement by applying a small vertical displacement. However, it is important to note that there are examples of lidar-satellite temperature measurements which cannot be brought into

good agreement with small vertical displacements. Two such examples can be found in Fig. 9. These
300  examples of poor agreement are almost exclusively found in winter on nights where the stratopause
is greatly disturbed.

[Figure]

Figure 7: The upper panel shows a case where the lidar and SABER were well aligned in altitude.
The lower panel shows a case where a vertical displacement of the SABER profile ameliorated the
agreement with the lidar measurement.

[Figure]

Figure 8: The upper panel shows a case where the lidar and MLS were well aligned in altitude. The lower panel shows a case where a vertical displacement of the MLS profile ameliorated the agreement with the lidar measurement.

[Figure]

Figure 9: Two examples of poor matches between lidar and satellite temperature profiles (MLS upper panel, SABER lower panel). These mismatches mainly occur between late November and early April on nights where the stratosphere was disturbed and experiencing a warming.

**5.2  Trends in Vertical Offset between Lidar and Satellites**

We calculated an offset for every coincident measurement between the  lidars and SABER and the  lidars and MLS. The monthly average of this altitude offset value is represented in Fig. 305 10 as a  green line for years where the comparisons were primarily between LTA and the satellites and as a blue line for years where $LiO_3S$ temperatures were used. The green and blue shaded regions are the respective standard deviations. Given the reduced vertical resolution of the temperature profiles from $LiO_3S$, the least-squares minimized correction for stratopause height is less sensitive to small and medium scale fluctuations

310 in the temperature profiles such as the triple peak structure seen in the lower panel of Fig. 7. As a result, comparisons between LiO₃S and both satellites (blue curve in Fig. 10) tend toward the mean altitude displacement. This effect is more pronounced when comparing with SABER, which has a finer vertical resolution, than when comparing with MLS which has a coarser vertical resolution. There is a clear, but imperfect, seasonality to these altitude displacements.

[Figure]

Figure 10: The upper panel features the monthly average displacement of SABER measurements with respect to the OHP lidars (green for LTA and blue for LiO₃S). The standard deviation is given as the shaded area. The mean offset (magenta) is 1446 m with a standard error of 49 m. The lower panel shows the same analysis with the monthly average MLS displacement. The mean value is 911 m with a standard error of 90 m.

315 Superimposing the traces shown in Fig. 10 onto the colour plots in Fig. 3 and Fig. 5 shows a clear correlation between lidar-satellite temperature anomalies and mean monthly altitude displacement between the lidar and satellite temperature profiles, as shown in Fig. 11.

[Figure]

Figure 11: The upper panel features the monthly median temperature differences between the lidar and  MLS seen in 5 with the estimated vertical displacement of the stratopause height overplotted. The lower panel features the monthly median temperature differences between the lidar and  SABER seen in 3 with the estimated vertical displacement of the stratopause height overplotted. The black line represents comparisons between LTA and the satellite and the grey line represents comparisons between $LiO_3S$ and the satellite.

**6 Recalculated Lidar-Satellite Temperature Differences**

We have attempted to make a more accurate comparison of the lidar and satellite temperatures by using the stratopause height as a common altitude reference. We re-calculated the lidar-satellite temperature differences shown in Fig. 4 and Fig. 6 after displacing the satellite measurement by a scalar value. Each satellite measurement was shifted vertically according to the lidar derived stratospheric displacements shown in Fig. 10.

[Figure]

Figure 12: Corrected seasonal temperature differences between the lidar and the vertically displaced SABER temperatures. The  magnitude of the temperature differences is reduced  in both the stratosphere and mesosphere over the majority of the altitude range when compared to a similar uncorrected temperature difference contour seen in Fig. 4 .

In Fig. 12 we see that by displacing the SABER temperature profiles so that the stratopause height is the same in both the lidar and satellite measurements we have  reduced the maximum winter time stratospheric warm bias  from approximately 8 K to 4 K. The summer time mesospheric cold bias of -10 K has likewise been reduced by between 4 and 6 K depending on altitude and season. The remaining bias in both the stratosphere and mesosphere cannot be further minimized by a simple vertical shift. The altitude dependent correction which would be required to correct the temperature lapse rate is beyond the scope of this work.

[Figure]

Figure 13: Corrected seasonal temperature differences between the lidar and the vertically displaced MLS temperatures. The structured nature of the temperature bias seen in Fig. 6 remains unchanged by the vertical correction.

In Fig. 13 we see that displacing the MLS temperature profiles was less successful than in the case of the SABER measurements. We have reduced the magnitude of beginning and end of winter time stratospheric warm bias by up to  5 K during the months of March, April, October, and November but the correction does not completely eliminate the issue. As well we have an improvement of  5 K in the biased layer at 65 km. However, the horizontal layering inherent in the MLS temperature data makes determining a scalar correction even more challenging than in the case of SABER.

We have replotted the seasonal ensemble temperature difference curves shown in the lower panel of Fig. 3 (lidar-SABER) and Fig. 5 (lidar-MLS) alongside the ensemble temperature differences after we applied the correction for stratopause height. Figure 14a shows the ensemble temperature difference for all 1741 lidar-MLS temperature comparisons before correction (red) and after correction (magenta). The prominent warm bias near 45 km has been reduced from 8 K to 6 K but the cold biases at 53 km, and 68 km are made worse by the correction. To understand this result we can look at the seasonal dependence of the applied correction. Figure 14c is the summer ensemble temperature difference (MJJA) consisting of 554 lidar-MLS temperature comparisons before correction (red) and after correction (magenta). There is marginal improvement after correction below 55 km but the change is not significant at $2\sigma$ and the structure of the temperature bias remains unchanged. Figure 14e is the winter ensemble temperature difference (NDJF) consisting of 653 lidar-MLS temperature comparisons before correction (blue) and after correction (magenta). There is significant improvement of 4 K in the large cold bias at 45 km. The corrected lidar-MLS comparison is also significantly worse near the cold bias at 63 km.

Figure 14b shows the ensemble temperature difference for all 1100 lidar-SABER temperature comparisons before correction (blue) and after correction (magenta). the stratopause height correction has reduced the stratospheric warm bias from 4 K to less than 1 K and has reduced the mesospheric cold bias from -4 K to -1 K. The warm bias above 70 km has been slightly increased. Figure 14d is the summer ensemble temperature difference (MJJA) consisting of 306 lidar-SABER temperature comparisons before correction (red) and after correction (magenta). There is a significant 3 K reduction in the warm bias at 45 km and a significant reduction in the mesospheric cold bias from -6 K to -3 K. Figure 14f is the winter ensemble temperature difference (NDJF) consisting of 397 lidar-SABER temperature comparisons before correction (blue) and after correction (magenta). By applying the altitude correction we have eliminated the 'S' shape in the temperature difference curve between 30 and 60 km. There is a significant 1 K constant warm bias that remains after correction. Above 70 km there is no statistically significant change.

[Figure]

[Figure]

(a) Median temperature difference for 1741 lidar minus MLS temperature profiles from 2004 to 2018. Red is the original ensemble and magenta is the ensemble after correction.

(b) Median temperature difference for 1100 lidar minus SABER temperature profiles from 2002 to 2018. Blue is the original ensemble and magenta is the ensemble after correction.

[Figure]

[Figure]

(c) Median summer (MJJA) temperature difference for 554 lidar minus MLS temperature profiles from 2004 to 2018. Red is the original ensemble and magenta is the ensemble after correction.

(d) Median summer (MJJA) temperature difference for 306 lidar minus SABER temperature profiles from 2002 to 2018. Red is the original ensemble and magenta is the ensemble after correction.

[Figure]

[Figure]

(e) Median winter (NDJF) temperature difference for 653 lidar minus MLS temperature profiles from 2004 to 2018. Blue is the original ensemble and magenta is the ensemble after correction.

(f) Median winter (NDJF) temperature difference for 397 lidar minus SABER temperature profiles from 2002 to 2018. Blue is the original ensemble and magenta is the ensemble after correction.

Figure 14: Ensemble for lidar minus MLS temperature differences (left) and lidar minus SABER (right). Ensembles for all profiles are on the top row, summer (MJJA) profiles in the middle row, and winter (NDJF) in the bottom row.

**7   Discussion**

365 ### 7.1   The need for vertical altitude correction of satellite data

 Improved observations of stratospheric and mesospheric temperature profiles and dynamical phenomena are required to advance our understanding of the middle atmosphere. The process of ground to satellite measurement comparison and validation is a vital ongoing scientific activity. By comparing long term, stable, continuous, high quality temperature measurements, such

370 as those made by the lidars at OHP, to other datasets we can help to identify potential issues with calibration or retrieval algorithms.

We have presented individual cases in Fig. 7 and Fig. 8 where both MLS and SABER temperature profiles benefited from a slight vertical displacement based on lidar derived stratopause height. While this scalar adjustment does not correct for non-linear distortions in the altitude vector it can

375 significantly reduce the magnitude of the temperature bias in the stratosphere and lower mesosphere  as seen in Fig. 14a and Fig. 14b. This technique does not seem to work well when the stratopause is highly disturbed as can be seen in the two winter time examples in Fig. 9. The implications of satellite underestimation of sudden stratospheric warming events is of particular concern to reanalysis projects attempting to model middle atmosphere dynamics. However, by using lidar data to

380 supplement the satellite record these fast dynamical processes can be better resolved.

**7.2   Temperature biases between OHP lidar and SABER**

In the companion publication (Wing et al., 2018a) we attempted to reduce the magnitude of the initialization induced lidar warm bias which is often reported above 70 km. We have reduced the bias by up to 5 K near 85 km and nearly 20 K at 90 km. There still remains some residual systematic

385 warm bias between the lidar satellite comparisons in this publication.

The average $9.9 \pm 9.7$ K bias at 80 km reported by (Dawkins et al., 2018) using 9 different metal layer resonance lidars compares favorably to our ensemble bias of 5 K at 80 km Fig. 14b. Given that the resonance lidars do not initialize their temperatures using the same inversion algorithm as the Rayleigh lidars, and that the resonance lidars have a minimum uncertainty near 85 km, perhaps our

390 Rayleigh temperatures are not as influenced by a priori as we thought. Further work needs to be done on the  topic of initialization related bias to fully address the effects of noise and a priori choice on high altitude Rayleigh lidar retrievals. However, we are encouraged by our results and cannot discount the possibility that some of the remaining temperature difference is due to incorrect altitudes in the satellite data product.

395 When considering the residual temperature differences between the OHP lidars and SABER after the altitude correction based on lidar derived stratopause height we can see that much of the seasonal variability in the stratosphere and mesosphere has been reduced. We are still left with a general  summer time cold bias over most of the atmospheric column, except near 45 km, which

now achieves a maximum of -4 K in the June mesosphere. We cannot explain this bias from the
perspective of the lidar data as nothing in our range resolution changes, our data acquisition cadence
and measurement duration are very similar (Wing et al., 2018a), and we are well into the linear region
of lidar count rates and are not influenced by our a priori or saturated count rates. It is possible that
there could be a tidal contribution as  summer time lidar measurements start a bit later
than  winter time measurements due to a shorter astronomical night. However, given that
our criteria for coincidence were chosen to minimize the effects of the first few tidal harmonics this
seems unlikely. It is also possible that there is a seasonally dependent bias in the a priori used in the
satellite retrieval of the geopotential height which could influence the satellite altitude vector.

The cold bias seen below 30 km is most likely due to possible contamination in the lidar data
from aerosols and saturation in the low gain Rayleigh channel. Current OHP lidar measurements
use Raman scatter data to correct for these effects and produce temperature profiles down to 5 km.
However, this Raman data is not available for the entire 2002 to  2018 analysis period so we
have opted not to include it in this work.

**7.3 Temperature biases between OHP lidar and MLS**

As with the comparison between the lidar and SABER, the lidar and MLS comparison has a pro-
nounced warm bias above 70 km which is in keeping with previous studies. However, the magnitude
and extent of this warm bias in MLS is much more pronounced than in the SABER comparison plot.
Much of this difference is due to the reduced vertical resolution of MLS at these high altitudes. This
holds true particularly when comparing lower vertical resolution lidar data to MLS.

The lidar MLS comparison has a  winter time stratospheric warm bias which is not
much reduced by simply shifting the location of the MLS stratopause  (Fig. 14e). We have reduced
the magnitude of the difference by 4 K but the stratopause altitude correction was markedly less
successful than in the case with SABER. It is almost universally the case that sudden stratospheric
warmings seen by the lidar are missed or smoothed over in the corresponding MLS measurement.
Figure 9 (upper panel) is very much a typical comparison for periods when the stratosphere is highly
disturbed. There is a limit to how much can be done to improve the lidar-MLS comparison using a
simple scalar correction.

The vertical structure which dominates much of the middle portion of the lidar MLS comparison is
also difficult to account for. The structure is particularly evident in Fig. 14c and is nearly insensitive
to our applied altitude correction. There is nothing in the lidar technique that could  explain
this pattern.
 A similar horizontal banding pattern is seen in the comparison of MLS
to The European Center for Medium-range Weather Forecasts (ECMWF) assimilation in the MLS
geopotential validation paper (Schwartz et al., 2008). The effect is most likely an artefact introduced
in some stage of the satellite retrieval. Studies like ours provide a perfect opportunity to incorporate lidar information into the satellite retrieval and improve the satellite data products. Given the confidence we have in the fixed width and amplitude of the vertical kernels in the lidar measurement, a lidar altitude and temperature vector could be used to recalculate the MLS  geopotential and temperature profiles to help identify the source of this artefact.

It is also important to acknowledge that simply correcting for stratopause height offset was counter productive for our lidar-MLS comparisons above 50 km as seen in Fig. 14a. It is likely that any potential lidar-derived correction for MLS will be more complex than a simple scalar offset. Such a correction may even have different functional forms in the stratosphere and mesosphere.

**7.4 Seasonality of Temperature biases between OHP lidar and satellites**

We have seen the 5 K difference between lidar-SABER stratopause temperatures which was reported in (Sivakumar et al., 2011) however, unlike this study we have found a clear seasonal dependence. We have correlated this temperature bias directly to a vertical displacement of the satellite altitude with respect to the lidar altitude and not with the Annual Oscillation. Further work must be done to explore the possibility of North Atlantic Oscillation/Annual Oscillation effects but a quick correlation of relative vertical displacement seen in Fig. 10 and a monthly average AO phase shows an R squared value of only 0.04 for SABER and 0.03 for MLS. There are isolated periods of up to a year where it seems like the correlations are significant however, it is clear that over a period of nearly a decade the AO phase and winter time stratospheric temperature anomaly are not correlated. The 5 K stratospheric warm bias was attributed to tides in (Yue et al., 2014) however, this explanation cannot explain the seasonal nature of this bias found in this work nor explain why a simple vertical displacement of the satellite stratopause height offers a suitable correction.

**8   Conclusions**

We can draw the following conclusions from the comparison of the lidar and satellite temperature measurements.

**1)** We have found the same systematic 5-15 K warm bias in the lidar-satellite comparisons above 70 km found in studies like (García-Comas et al., 2014), (Taori et al., 2011), (Taori et al., 2012b), (Taori et al., 2012a),(Dou et al., 2009), (Remsberg et al., 2008), (Yue et al., 2014), (Dawkins et al., 2018) , and (Sivakumar et al., 2011). We have attempted to carefully account for the the background-induced warm bias in high altitude Rayleigh lidar temperatures. We believe that the algorithm set out in the companion publication (Wing et al., 2018a) is robust and accounts for many of the uncertainties in the lidar initialization process. However, we are as yet unable to determine to what extent the a priori estimate warms the lidar temperature retrieval at these heights.

**2)** We have seen a layered summer stratosphere-mesosphere cold bias in lidar-MLS seasonal temperature comparisons with peak differences at 37 km, 50 km, and 65 km. There is nothing in the lidar

data or retrieval algorithm which could account for this structure. The results of this study will be
470    useful for any future satellite validation studies in the style of (Schwartz et al., 2008) where lidar
data could be used as a reference dataset. In particular, lidar - satellite bias study results are useful
for the ongoing NASA project "The Mesospheric and Upper Stratospheric Temperature and Related
Datasets" (MUSTARD) which seeks to merge historic and ongoing satellite datasets.

**3)** The persistent summertime cold bias between the lidar and SABER results from a disagree-
475    ment in the thermal lapse rate above and below the stratopause which is independent of the scalar
stratopause height offset. Given that lapse rate is a fundamental geophysical parameter further work
must be done to explore possible errors in vertical resolution and altitude definition.

**4)** The periods of greatest lidar-satellite temperature disagreement are  found during times
when the middle atmosphere is highly disturbed. In particular, the amplitude of stratospheric warm-
480    ing events can be underestimated and features like double stratopauses can be missed in the satellite
measurements.

We have shown that ground based lidars can provide  reliable and consistent
temperature measurements over decades. This kind of high vertical resolution temperature database
is useful both as a validation source for other instruments as well as for fundamental geophysical
485    research.

*Acknowledgements.* The data used in this publication were obtained as part of the Network for the Detection of
Atmospheric Composition Change (NDACC) and are publicly available (see http://www.ndacc.org, http://cds-
espri.ipsl.fr/NDACC) as well as from the SABER (see ftp://saber.gats-inc.com) and MLS (see https://mls.jpl.nasa.gov)
data centres for the public access via their websites. This work is supported by the project Atmospheric dynam-
490    ics Research InfraStructure Project (ARISE 2) funded by funded by the European Union's Horizon 2020 re-
search and innovation programme under grant agreement No. 653980 French NDACC activities are supported
by Institut National des Sciences de l'Univers/Centre National de la Recherche Scientifique (INSU/CNRS),
Université de Versailles Saint-Quentin-en-Yvelines (UVSQ), and Centre National d'Études Spatiales (CNES).
The authors would also like to thank the technicians at La Station Géophysique Gérard Mégie at OHP .